# Development of a C6 Glioma Cell Model System to Assess Effects of Cathodic Passively Balanced Electrical Stimulation on Responses to Neurotransmitters: Implications for Modulation of Intracellular Nitric Oxide, Chloride, and Calcium Ions

**DOI:** 10.3390/brainsci12111504

**Published:** 2022-11-06

**Authors:** David C. Platt, C. Fiore Apuzzo, Marjorie A. Jones, David L. Cedeno, Ricardo Vallejo

**Affiliations:** 1Department of Chemistry, Illinois State University, Normal, IL 61790, USA; 2Division of Research, SGX Medical LLC, Bloomington, IL 61701, USA

**Keywords:** electrical stimulation, glioma cells, nitric oxide, glutamate, adenosine, D-serine, γ-aminobutyric acid

## Abstract

This research focused on the development of an astrocyte cell model system (C6 glioma) for the assessment of molecular changes in response to cathodic passively balanced pulsed electrical stimulation at a rate of 50 Hz (60 µs duration, 0.15 mA intensity). Cells treated with selected neurotransmitters (glutamate, adenosine, D-serine, and γ-aminobutyric acid) were monitored (using specific fluorescent probes) for changes in levels of intracellular nitric oxide, calcium ions, and/or chloride. ES exerted an inhibitory effect on NO, increased calcium and had no effect on chloride. Using this model, cells can be assessed qualitatively and quantitatively for changes and these changes can be correlated with the putative molecular effects that electrical stimulation has on astrocytes and their role in glia-mediated diseases. This model system allows for faster and cheaper experiments than those involving animal models due to the potential to easily vary the conditions, reduce the number of variables (especially problematic in animal models), and closely monitor the cellular effects.

## 1. Introduction

The electrical stimulation (ES) of neural structures in the central nervous system has been used as an alternative for pharmacological management of certain diseases, including pain, due to the limited response and significant side effects of such a conventional management. Unfortunately, cellular, and molecular mechanisms to explain the observed clinical benefits of this approach are still not well understood [1,2,3,4].

The use of animal models has served to better understand the mechanism of action of ES on relevant biological processes affected by neuroinflammation, such as chronic pain in which our group has an interest. While previous understanding of mechanisms focused on the effects of ES on neuronal depolarization, a new methodology, based on transcriptomic and proteomic analyses, supports the effects of electrical signals beyond neurons involving glial cells and neuroinflammation [5,6,7,8]. We previously used C6 glioma cells in culture subjected to ES as a model of astrocytes [9,10,11] and showed modulation of gene expression that was dependent on the anodic content of the stimulating biphasic pulses [9]. This work indicated that the effects on gene expression of a selection of glia-related genes were dependent on the ES waveform composition. We have now extended these studies to assess direct ES modulation of specific molecules and ions, using fluorescent probes, with and without the addition of select neurotransmitters (glutamate, γ-aminobutyric acid, adenosine, and D-serine).

Nitric oxide (NO) is a canonical participant in a myriad of signaling cascades. Most notably, NO facilitates vascular dilation, but it also participates in other physiological processes including stress responses [12,13,14]. Pertinent to our work, NO acts as a neurotransmitter in the nociceptive process and contributes to the development of central sensitization. Interestingly, NO may also participate in nociceptive inhibition and mediates the analgesic effects of pharmacological agents [15]. NO demonstrates exceptional reactivity albeit with a very brief half-life. As an endocrine signaling agent, these two properties are critical for eliciting a potent localized response quickly while preventing off target activation of other NO influenced processes farther away.

Glutamate (Glu) is an excitatory neurotransmitter which acts as a neuronal-glial signaling agent activating neurons and glial cells via one of ionotropic and metabotropic glutamate receptors [16]. These receptors influence changes in Ca^2+^ levels in the presence of glutamate. In astrocytes, glutamate signaling results in the release of Ca^2+^ from the endoplasmic reticulum into the cytoplasm and propagation of Ca^2+^ waves through gap junctions. This process leads to NO synthase activation and production of NO and other reactive nitrogen species [17,18]. In this work, glutamate was introduced to perturb the cells and modulate the intracellular release of chloride, Ca^2+^ and production of NO.

Chloride (Cl^−^) is the major extracellular anion, playing important roles in osmotic regulation and healthy neuronal functioning. Cl^−^ concentrations across neuronal membranes aid in the signaling function of γ-amino butyric acid (GABA). When intracellular Cl^−^ concentrations are low, GABA can hyperpolarize neurons and function as an inhibitory neurotransmitter. When intracellular Cl^−^ concentrations are high, GABA acts in an excitatory manner. The mobilization of Cl^−^ by glial cells can directly influence this balance by altering the extracellular Cl^−^ concentration in surrounding neurons, altering neuronal response to neurotransmitters [19].

Adenosine is an inhibitory neurotransmitter produced by phosphatase activity on adenosine 5′-triphosphate (ATP). In response to stressful stimuli and inflammation, ATP is released into the extracellular milieu where ectonucleotidases and tissue non-specific alkaline phosphatases, dephosphorylate it to yield excess levels of extracellular adenosine, which acts as a neurotransmitter binding to adenosine receptors [20,21]. Adenosine can also be produced intracellularly and transported extracellularly to act on an adenosine receptor. Degradation of the adenosine signal is accomplished in the cytosol by phosphorylation to AMP or deamination to inosine. The activity of ecto-adenosine deaminase or nucleoside transporter activity will return the extracellular adenosine level to its resting concentration. Adenosine is reported to cause NO production in microglial cells [22].

GABA is an endogenous neuromodulator that participates in numerous capacities within the central nervous system [23]. For instance, injection of interleukin-1 beta into rat forebrains induced morphological changes in glial cells resulting from NO production, causing the release of GABA [24]. Here, we use GABA to alter the production of NO.

D-serine is a coagonist of the N-methyl-D-aspartic acid (NMDA) receptor and participates in the activation of astrocytes leading to the development of mechanical hypersensitivity following peripheral nerve injury by increasing Ca^2+^ inflow [25]. Serine racemase, responsible for interconversion of the L- and D-isomers, is highly expressed in astrocytes and participates in neuro-glial transmission. In this work, D-serine was introduced to perturb the cells while monitoring the production of NO.

This study was implemented to develop a simple and reproducible in vitro method using an astrocyte model to gather data concerning the potential effects of ES on the production and maintenance of specific chemical species involved in the development and maintenance of neuroinflammation. We varied the order of ES relative to addition of a neurotransmitter or a control (phosphate-buffered saline (PBS)) to assess time dependent effects. We also evaluated the ability to monitor effects following additions in the presence of a single or dual fluorescent probes. Extrapolation of acquired in vitro results can serve to generate working hypotheses that can be evaluated in animal models, which should lead us to a better understanding of the effects of ES on neural tissue. In this publication, we demonstrate that careful selection of neurotransmitters and fluorescent probes gives insight into the cellular response to electrical stimulation making this a reliable in vitro model system.

## 2. Materials and Methods

### 2.1. Cell Culturing

Axenic *Rattus norvegicus* C6 glioma cells purchased from the American Type Culture Collection, Manassas, VA, USA (ATCC CCL-107) were grown in sterile 6-well plates using high glucose Dulbecco’s Modified Eagle’s medium (DMEM; Sigma Life Sciences D6429; St. Louis, MO, USA) supplemented with 7.5% (*v*/*v*) horse serum (ATCC; Manassas, VA, USA) and 3% (*v*/*v*) heat-treated fetal bovine serum (GIBCO; Waltham, MA, USA). This sera-supplemented medium was designated “complete medium”, whereas unsupplemented medium was designated as “incomplete”. Cells were grown at 37 °C under a 5% CO_2_ atmosphere and 90% humidity. To transfer adherent cells, trypsin (Sigma Life Sciences T4049; St. Louis, MO, USA) was used to release cells from the bottom of the plate in accordance with the manufacturer’s instructions. Trypsin was subsequently neutralized upon addition of complete medium by endogenous α-1 antitrypsin present. This cell preparation was then centrifuged (Labnet Hermle Z 400K, Edison, NJ, USA) at 2000 rpm for 10 min at 7 °C. The supernatant was discarded, and the resultant cell pellet was re-suspended in complete medium and plated as required. Large cultures of cells were maintained in CELLSTAR^®^ TC sterile 6-well plates (Greiner Bio-One, St. Louis, MO, USA). For experimentation, cells were harvested, as previously described, re-suspended in complete medium, and counted with a Handheld Automated Cell Counter (Millipore Scepter 2.0, Burlington, MA, USA) with a 60 μm sensor. After dilution to 3.0 × 10^5^ cells/mL, 150 μL of this cell suspension was applied to sterile 24-well cell culture plates (Falcon^®^, Fisher Scientific, Waltham, MA, USA) containing 150 μL of complete medium, yielding a total volume of 300 μL and 4.5 × 10^4^ cells per well. Experiments began when cells in the 24-well plates were deemed confluent on the second or third day in culture. All cell culturing was performed in a UV sterilized hood (Thermo Electron Corporation Forma Class II Biological Safety Cabinet, Waltham, MA, USA) for maintaining sterile conditions. Surfaces contacted within the hood were cleaned with 70% ethanol.

### 2.2. Fluorescent Probes and Neurotransmitter Preparations

Glutamate, adenosine, D-serine, and GABA were obtained from Sigma Aldrich (St. Louis, MO, USA) and used as provided, at biologically relevant concentrations (Table 1). Solids were dissolved in NanoPure water to prepare a stock solution. This solution was loaded into a syringe and sterilized with a 0.22 μm syringe driven filter (MILLEX^®^ GP, Millipore, Cork, Ireland). Aliquots were stored at −20 °C and thawed immediately before use.

2′-7′-dichlorofluorescin diacetate (DAF-FM; excitation/emission: 495/515 nm) was obtained from EDM Millipore Corp. (Burlington, MA, USA), prepared according to manufacturer’s instructions, and implemented with a modified procedure of the manufacturer’s directions. DAF-FM fluoresces in the presence of intracellular NO [26,27].

N-(Ethoxycarbonylmethyl)-6-methoxyquinolinium bromide (MQAE; excitation/emission: 350/460 nm) was obtained from Invitrogen (Waltham, MA, USA), prepared according to the manufacturer’s instructions, and implemented with a modified procedure of the manufacturer’s directions. The fluorescent signal from MQAE is quenched in the presence of intracellular Cl^−^ [28].

Fluo-4-AM (Excitation/emission: 494/506 nm) was obtained from ThermoFisher (Waltham, MA, USA), prepared according to the manufacturer’s instructions, and implemented with a modified procedure of the manufacturer’s directions. Fluo-4-AM fluoresces in the presence of intracellular calcium ions [29].

### 2.3. Electrical Stimulation Characteristics

Electrical stimulation (ES) was conducted in parallel culture wells using flat concentric bipolar electrodes (FHC Microelectrodes, Bowdoin, ME). Cathodic passively balanced pulses (60 μs width and 0.15 mA intensity) were applied at 50 Hz for 30 min to each well. ES parameters were selected to reflect those conventionally used in practice, while protecting the integrity of the electrode and cells [9]. A custom-built ES apparatus housed four electrodes, situating them such that the electrode tip is submerged in the medium, in separate wells simultaneously, but elevated above the bottom of the plate to which the glioma cells are adhered. ES occurred in 4 wells at a time, within the biosafety hood at room temperature (22–23 °C) in ambient atmosphere and in the dark.

Figure 1 and Figure 2 show the general experimental designs for the fluorescence experiments and the timing of neurotransmitter addition relative to time of ES. Figure 1 outlines the ES time dependency.

Figure 2 outlines the ES time dependency relative to addition of probe only or probe and neurotransmitter additions.

For time-dependent ES experimentation (Figure 1), a first set of wells were stimulated for 30 min, followed by replacing the medium in all the wells with probe-containing incomplete medium (DAF-FM or Fluo-4 AM). Select neurotransmitters (Table 1) or sterile PBS as control, were introduced to all wells at this point. After adding probes and neurotransmitters, ES was immediately continued on a second set of wells for 30 min. Following this, a third set of wells, to which the probes and neurotransmitters were added 30 min prior to ES, were then stimulated for 30 min. The plate was allowed to incubate for an hour following the third and final round of stimulation in the incubator at 37 °C and 5% CO_2_. Subsequently, medium was removed, and cells were washed (three times) with sterile 37 °C PBS with a final addition of warm PBS before fluorescent microscopy and spectroscopy.

For experimentation with co-incubated probes (Figure 2), supernatant medium was pulled off, discarded, and replaced with a hypotonic MQAE probe solution incubated according to the manufacturer, and then replaced with either NO probe (DAF-FM) or Ca^2+^ probe (Fluo-4 AM) with or without neurotransmitter in incomplete medium. DAF-FM and Fluo-4 AM probes were not tested together due to overlapping excitation/emission bands. One set of wells was then subjected to ES for 30 min. The neurotransmitters Glu, GABA or adenosine were also added to a second set of wells in which probes were co-incubated and let incubate for 30 min without ES. Following incubation, medium was removed and discarded. Cells were washed three times with sterile 37 °C PBS with a final addition of warm PBS before fluorescence measurements.

### 2.4. Fluorescence Measurements

Fluorescence microscopy was performed at 20× magnification using an epifluorescence microscope (Keyence BZ-X810, Itasca, IL, USA) fitted with GFP and DAPI filter cubes. Cells in wells were imaged by first imaging the control cells (no ES, no neurotransmitter) and setting the light exposure and scope recording values to the levels used for optimal fluorescence acquisition in the control well. Brightfield and fluorescent images were overlaid using Keyence analyzer software and cells were counted by hand. Three representative images were counted for each individual condition. Increased green fluorescence represents either DAF-FM (NO) or Fluo-4 AM (Ca^2+^), whereas decreased blue fluorescence represents MQAE (Cl^−^). Cyan color was artificially input using the analyzing software wherever green and blue fluorescence overlapped.

Fluorescence spectroscopy measurements were then obtained following image acquisition. Supernatant PBS was discarded and 200 μL of 10% (*w*/*v*) sodium dodecyl sulfate (SDS) in water was added to solubilize cells. Once cells were solubilized, samples were placed in a fluorescence cuvette for analysis using a fluorescence spectrometer (LS-55, Perkin-Elmer; Waltham, MA, USA). Excitation and emission wavelengths were set for the appropriate probes and fluorescence collected every second for 60 s for each well. Fluorescence intensities per well are reported as averages with standard deviations. A one-way analysis of variance (ANOVA) with post hoc Tukey’s test was performed on experimental data sets grouped by condition to determine statistical significance of difference in means (*p* < 0.05).

### 2.5. Data Analysis

Results from time-dependent experiments are reported in bar graphs using three distinct terms when referring to electrically stimulated samples. Cells stimulated before the addition of probe and neurotransmitter are referred to as “ES before …”, whereas cells stimulated immediately after addition of these are labeled as “ES and …”. The third term is referred to as “… before ES” to denote wells that were electrically stimulated 30 min after addition of probe and neurotransmitter (see Figure 1). All electrically stimulated wells are denoted in green on graphs whereas same addition, but not electrically stimulated cells are denoted in blue, and control cells only (PBS addition and no ES) is indicated in black. Mean fluorescence intensities from cells co-incubated with Cl^−^ probe and either NO or Ca^2+^ probe are reported in bar graphs. Significant differences within neurotransmitter groups in time-dependent experiments and between groups were determined by ANOVA with Tukey post hoc test (*p* < 0.05).

## 3. Results

### 3.1. Time-Dependent Probing of NO or Ca^2+^ after Cell Incubation with Neurotransmitters

Initially, the NO probe, DAF-FM, was used to determine the appropriate methodology for incubation and data acquisition. The percentage of fluorescent cells was obtained by total cell counts using brightfield microscopy and counts of cells exhibiting fluorescence (Figure 3). The presence of GABA provided the highest percent of fluorescent cells when stimulated 30-min post-addition of the probe and neurotransmitter. Other GABA-based conditions also provided relatively high fluorescent signals when compared to the control cells (black bar) as well as the other experimental conditions. Adenosine addition before ES shows the highest percent of fluorescent cells among adenosine samples; however, it varies little from the control cells. Electrical stimulation before or concomitant with adenosine addition reduced the percent of cells exhibiting the signal for nitric oxide.

Many of the conditions have a low percentage of counted fluorescent cells (0–10%), although these resulted in larger values when fluorescence was measured from solutions (from lysed cells counted in Figure 3A) using spectroscopy (Figure 3B). This implies that fluorescent microscopy may underestimate level of detectable probes since the images are captured with the exposure values determined by control cells exposure value. Figure 4 shows that upon normalization of measurements to control, cell counts from fluorescence images correlate with spectroscopic measurements in a non-linear manner. Therefore, subsequent quantitative analyses used the spectroscopic measurements. The addition of adenosine and GABA significantly increased (*p* < 0.001) NO probe fluorescence relative to control cells, while the addition of Glu significantly decreased it (Figure 3B). In all cases, ES before or right after addition of the neurotransmitter significantly lowered (*p* < 0.001) fluorescence intensities, while 30-min incubation of the neurotransmitter previous to ES resulted in a significant increase in NO probe fluorescence (GABA, adenosine), similar fluorescence (Glu), or a significant decrease (D-serine) relative to no ES.

Fluorescence spectroscopic measurements were also obtained for cells incubated with the intracellular Ca^2+^ probe (Fluo4-AM) in the presence of each of the various neurotransmitters with or without ES (Figure 5). The addition of neurotransmitters significantly decreased (*p* < 0.001) Ca^2+^ probe fluorescence relative to control cells. Incubation of cells with D-serine, adenosine, or GABA for additional 30 min before ES significantly increased Ca^2+^ probe fluorescence intensities relative to those without ES. Interestingly, 30-min incubation with Glu followed by ES did not show any difference relative to no ES. ES before or right after the addition of D-serine or adenosine significantly increased (*p* < 0.001) fluorescence intensities, while addition of GABA or Glu significantly decreased them.

### 3.2. Simultaneous Effects of ES or Neurotransmitters on Intracellular Species

Concomitant testing of MQAE and DAF-FM or Fluo-4 AM (as described in Figure 2) was pursued to get a better view of the relationships specific neurotransmitters have between each other in the presence of ES. Cell image overlays (Figure 6) provide notable contrast between individual conditions, where cells subjected to ES for 30 min have low indication of the presence of NO, while cells incubated with 250 nM adenosine have a significant amount of green and cyan fluorescence indicating the presence of high amounts of NO. Statistical analysis from the fluorescence spectroscopy measurements (Figure 7A) supports initial conclusions from image observation about DAF-FM fluorescence. Mean fluorescence intensities between groups are statistically significant (*p* < 0.05), except for GABA vs. glutamate. Samples subjected to ES have a much lower DAF-FM signal than any other group, agreeing with the first approximation from the microscopic imaging. The effect of incubation with either glutamate, adenosine or GABA is very similar and represents a significant increase relative to control cells.

MQAE fluorescence probing of Cl^−^ showed no statistical difference between experimental groups, although, addition of glutamate and GABA led to modest increases relative to control cells. Fluorescence intensities (Figure 7B) from MQAE are relatively low, compared to DAF-FM signal intensities, implying large amounts of Cl^−^ are retained in the cells since the probe is quenched in the present of the ion.

Coincubation of the Ca^2+^ (Fluo-4 AM) and Cl^−^ (MQAE) probes yielded similar results to that of the previous experiment. Cell overlays did not show the large differences as seen when incubating the Cl^−^ probe with the NO probe (DAF-FM), however, slight differences in color are noticeable (Figure 8). Control and electrically stimulated cells show nearly the same amount of green Fluo-4 AM fluorescence from the images.

Fluorescence intensity values highlight this similarity quantitatively (Figure 9A). There is no statistical difference in intracellular Ca^2+^ fluorescence between control cells (no ES, no neurotransmitter) and ES cells. All neurotransmitters induced statistically significant elevated levels of intracellular Ca^2+^ relative to control. Congruent with the previous observations when both NO and Cl^−^ were monitored in the same incubations, MQAE probing of Cl^−^ produced no statistical different results between experimental groups and control cells (Figure 9B) again indicating high levels of Cl^−^ in the cells.

## 4. Discussion

We have studied the effect of passively balanced cathodic pulses (50 Hz) on intracellular nitric oxide (NO), chloride (Cl^−^), and calcium ions (Ca^2+^) in C6 glioma cells using fluorescent probes in the presence of selected neurotransmitters. We used both fluoroscopic imaging and spectroscopic measurements to assess the effect of ES and neurotransmitters. Although fluoroscopic imaging allows for acquisition of high-quality images that assess changes in cell morphology in the presence of neurotransmitters, and/or after subjected to an ES, meaningful interpretations are enhanced when considered in conjunction with spectrophotometric measurements of the cell lysates. In general, trends in cell counts from fluorescence images correlate with spectroscopic measurements, although not in a linear manner. The spectroscopic methodology should be more reliable as it accounts for the treatment effect on all the cells in the sample and not a representative sample based on cell counting from the imaged samples. This method is particularly useful if the fluorescent signal of the probe is quenched by the chemical species (such for Cl^−^ in this study) which limits the sensitivity of the cell counting methods due to the inability to tease out any real differences due to low fluorescent signals. The method also provides a relatively easy and quick way to assay the effects due to ES on the levels of NO, Ca^2+^ and Cl^−^ in C6 glioma cells as an in vitro model of astrocytes prior to performing animal studies. The use of this model can aid researchers in acquiring preliminary data in a timely and cost-efficient fashion before conducting long-term and costly in vivo animal studies. Besides, primary astrocyte cells can only be obtained by explanting them from animals and cannot be passaged. Using the C6 glioma cell model, compounds of interest can be evaluated alone (such as monitoring NO with or without ES) or in combinations (as monitoring NO and Cl^−^ with and without ES) from the same cells. This provides a way to evaluate multifactorial effects.

When investigating the effects of ES and neurotransmitters in intracellular NO, Ca^2+^ and Cl^−^, we have used two approaches. In one of them, we looked at the dependency of the time in which ES was carried out relative to the presence of neurotransmitter (Figure 1). In the second approach we co-incubated two probes (Figure 2), while glioma cells were evaluated with and without added neurotransmitters and with or without ES. This proof-of-concept study provided interesting insight information regarding effects of ES on relevant processes involved in the interaction of neurotransmitters and the glioma cells. Known cellular effects of neurotransmitters such as Glu and adenosine on NO production glial cells including astrocytes [22,30,31] are reproduced in vitro using the glioma model cell system. Prolonged exposure to excess extracellular Glu and adenosine, as well as GABA and D-serine induced production of intracellular NO which is associated with receptor-mediated downregulation of the antioxidant intracellular mechanisms that protect cells. In all cases, application of cathodic ES (30 min, 0.15 mA, 60 µs pulses at 50 Hz) before or right after the addition of neurotransmitter resulted in lower detection of intracellular NO. Interestingly, ES also significantly reduced the amount of intracellular NO by around 40% relative to control unstimulated cells in the absence of neurotransmitters. Thus, in principle, ES could provide a reduction of NO induced by the presence of excess neurotransmitter. It was observed that the timing of the start of ES relative to the introduction of neurotransmitter seems relevant. When ES was started immediately after the addition of the neurotransmitter, NO levels were about 20–30% lower than NO levels found in unstimulated cells. However, when the neurotransmitter is allowed to incubate for 30 min before ES, the effect of ES is less pronounced. For GABA and adenosine, NO level is actually increased by ~10–15% relative to no ES, while for D-serine the reduction is about 15% and for Glu there is no difference. It is quite interesting that when cells are stimulated before incubating with Glu or D-serine, the fluorescence due to intracellular NO is the lowest. These observations suggest that ES may disrupt the chain of intracellular events that follow the interaction of the Glu/D-Serine/NMDA receptor that leads to intracellular Ca^2+^ influx and phosphorylation events involved in the activation of NO synthase and NO production.

It was observed that intracellular Ca^2+^ in the C6 glioma cells was increased following the prolonged incubation with adenosine, GABA or Glu. This is consistent with the effects of these neurotransmitters in astrocytes [16,32,33]. Intracellular Ca^2+^ in cells exposed to ES under our experimental condition in the absence of neurotransmitter was not affected relative to unstimulated control cells. ES increased intracellular levels of Ca^2+^. The timing of ES exposure relative to the introduction of the neurotransmitter seemed relevant for intracellular Ca^2+^ levels. Incubating cells for 30 min with GABA, adenosine or D-serine followed by 30-min ES, produced Ca^2+^-based levels that were about 20–40% larger than those observed in equivalent unstimulated cells. ES right after the addition of the neurotransmitter produced larger Ca^2+^ levels than unstimulated cells with adenosine and D-serine, and smaller levels with Glu. Stimulating the cells before adding neurotransmitter only produced larger Ca^2+^ values than unstimulated cells with D-serine. Previous work has shown that ES can induce an increase in intracellular calcium in astrocytes that when released form the cell induces calcium waves that propagates to other astrocytes [34,35]. Interestingly, we did not observe an increase in intracellular Ca^2+^ with ES without adding excess neurotransmitter, but the addition of neurotransmitter definitively enhanced the increase of intracellular calcium upon ES. Our experimental design does not allow recording of calcium waves that might be produced via ES.

Neither neurotransmitters nor ES had any effect on intracellular Cl^−^ concentration relative to control cells. It was also of interest that levels of Cl^−^ did not change to any significant extent in contrast of changes in the Ca ^2+^ or NO changes thus implying specificity.

Extension of the initial time-dependent experiment to the concomitant probe experiments is an important future direction for this work, as it is the use of confocal fluorescent microscopy imaging with adequate time-resolution that will allow the detection of intracellular calcium dynamics as a function of the ES parameters (pulse duration, intensity, frequency). Future work will also include the quantification the effect of ES parameters in gene expression levels.

## 5. Conclusions

This work described a methodology that allows for the study the effect of ES on intracellular species in C6 glial cells as an astrocyte in vitro model. Fluorescent from intracellular probes can provide quantifiable assessments via cell counting of imaged cells or spectrophotometric measurements from cells in a sample after detergent-mediated lysis and solubilization. Our results imply that spectroscopic measurements may be a preferred quantitative method of monitoring changes. One benefit to the described methodology is the unique ability that allows for experimental designs to systematically evaluate the dependence of the effect on the characteristic parameters of the ES signal (pulse width, frequency, recharge balance, and amplitude) For instance, this study showed that exposure of C6 glial cells to 30 min of a passively balanced cathodic pulsed signal of 60 µs duration and 0.15 mA intensity at a frequency of 50 Hz can inhibit the production of intracellular NO, which is increased by the incubation of cells with excess amounts of neurotransmitters. ES induced the increase of intracellular calcium levels in the presence of these neurotransmitters, while ES did not have effect on intracellular chloride levels. Ultimately, data obtained from this model system should be used to help guide in vivo animal studies and subsequent clinical applications to better understand the effects of ES in pain therapy and other neurodegenerative conditions in which glia cells may play a role.

## Figures and Tables

**Figure 1 brainsci-12-01504-f001:**
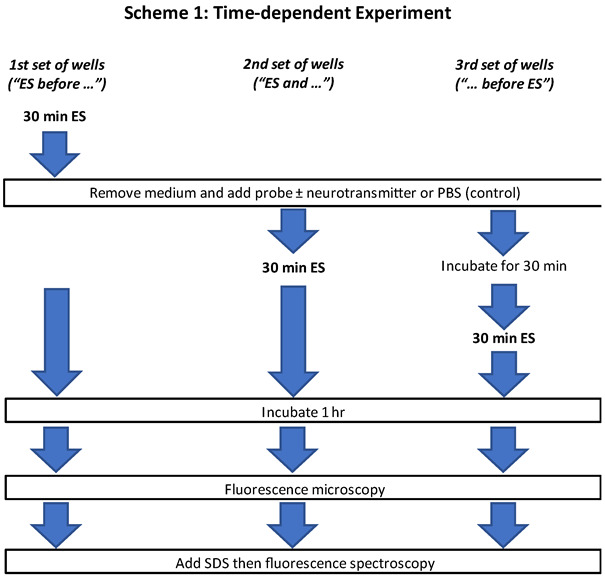
Time dependent experimental design varying the time of ES and neurotransmitter introduction. ES: electrical stimulation; PBS: phosphate buffered saline; SDS: sulfate dodecyl sodium.

**Figure 2 brainsci-12-01504-f002:**
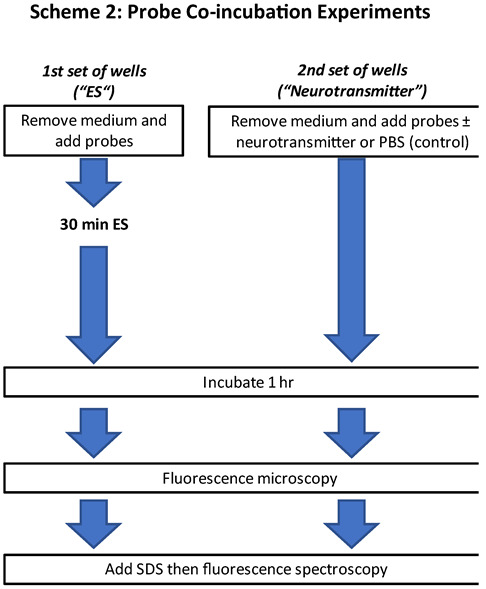
Experimental design introducing both MQAE and DAF-FM or Fluo4-Am as probe with and without neurotransmitter co-incubation. ES: electrical stimulation; PBS: phosphate buffered saline; SDS: sulfate dodecyl sodium.

**Figure 3 brainsci-12-01504-f003:**
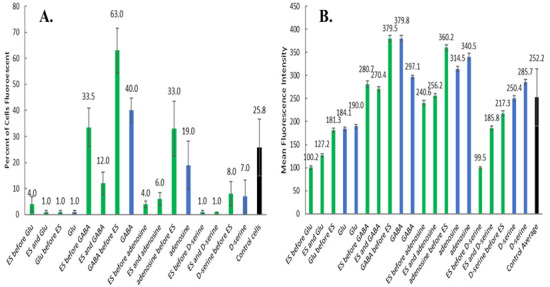
(**A**) Cell counts of cells incubated with NO probe (DAF-FM) at 3 different time points of ES and different neurotransmitters (Figure 1) Values are mean ± SD for *n* = 3 images per condition. (**B**) Fluorescence spectroscopy measurements of individual solubilized well solutions (from cells counted in Figure 1) incubated with NO probe (DAF-FM) and with or without ES and neurotransmitters (Figure 1) Values are mean ± SD for 60 measurements of a single well. Control Average is mean ± SD of 240 measurements from 4 replicate wells. Green bars indicate cells stimulated before or after additions, blue bars indicate cells with additions but no ES, and black bar indicates PBS control cells with no ES.

**Figure 4 brainsci-12-01504-f004:**
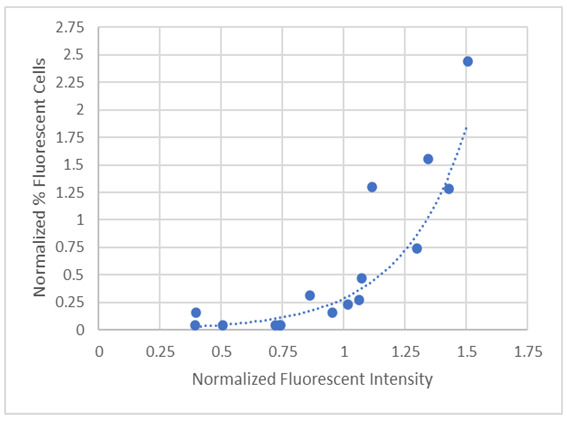
A graph showing the exponential correlation of the values from Figure 3A,B after normalizing to the control values showing a strong correlation of the determinations (R^2^ = 0.852, y = 0.007e^3.72x^).

**Figure 5 brainsci-12-01504-f005:**
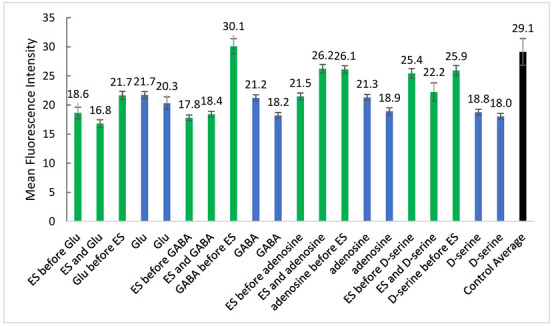
Fluorescence spectrometry measurements of individual solubilized cell solutions incubated with Ca^2+^ probe (Fluo4-AM) and with or without ES and neurotransmitters (Figure 1). Green bars indicate cells stimulated before or after additions, blue bars indicate cells with additions but no ES, and black bar indicates PBS control cells with no ES. Values are mean ± SD for 60 measurements of a single well. Control Average is mean ± SD of 240 measurements from 4 replicate wells.

**Figure 6 brainsci-12-01504-f006:**
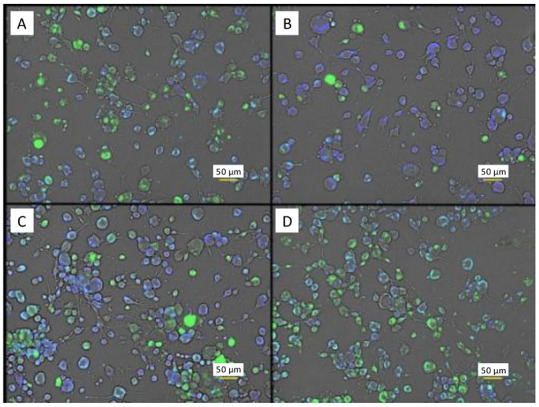
Fluorescence microscopy images of C6 glioma cells in the same plate in the presence of both 20 μM NO probe (DAF-FM, green) and 5 mM Cl^−^ probe (MQAE, blue). Cyan color denotes coexistence of intracellular NO and Cl^−^. (**A**) PBS control cells. (**B**) PBS cells after ES. (**C**) 10 mM L-glutamate. (**D**) 250 nM adenosine. Scale bars are 50 µm.

**Figure 7 brainsci-12-01504-f007:**
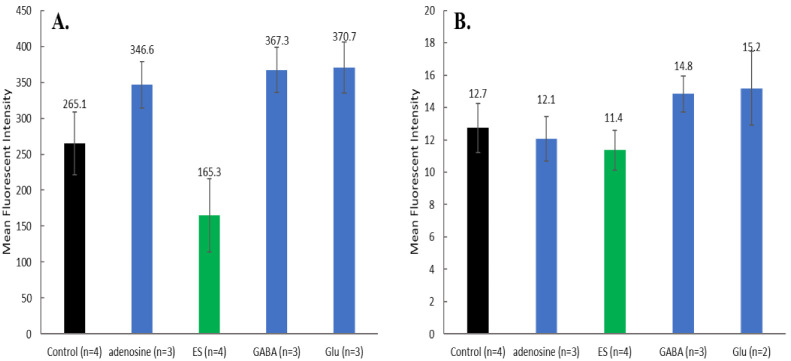
(**A**) Mean fluorescent intensity ± SD of DAF-FM (NO probe) averaged by condition group (Figure 2). All differences in mean intensities are significant except GABA vs. Glu by ANOVA with Tukey post hoc test (*p* < 0.05). (**B**) Mean fluorescent intensity ± SD of MQAE (Cl^−^ probe) averaged by condition group (Figure 2). No statistical differences were found between any of the groups.

**Figure 8 brainsci-12-01504-f008:**
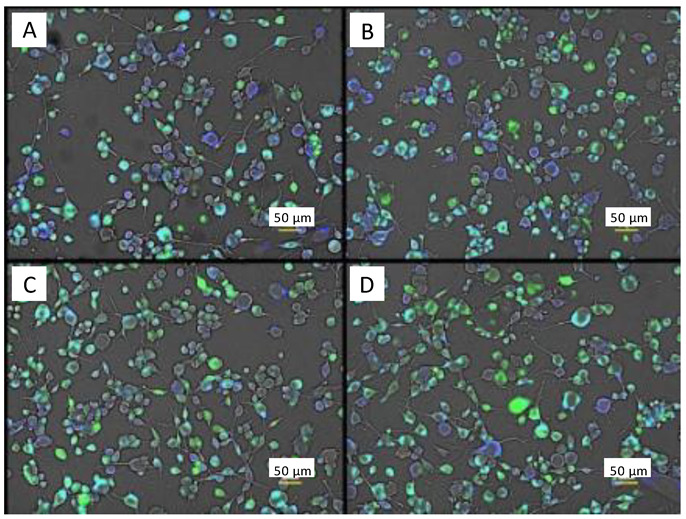
Fluorescence microscopy images of C6 glioma cells in the same plate in the presence of both 5 μM Ca^2+^ probe (Fluo-4 AM, green) and 5 mM Cl^−^ probe (MQAE, blue). Cyan denotes coexistence of Ca^2+^ and Cl^−^ in the cells. (**A**) PBS control cells. (**B**) PBS control cells after ES. (**C**) 10 mM L-glutamate. (**D**) 250 nM adenosine. Scale bars are 50 µm.

**Figure 9 brainsci-12-01504-f009:**
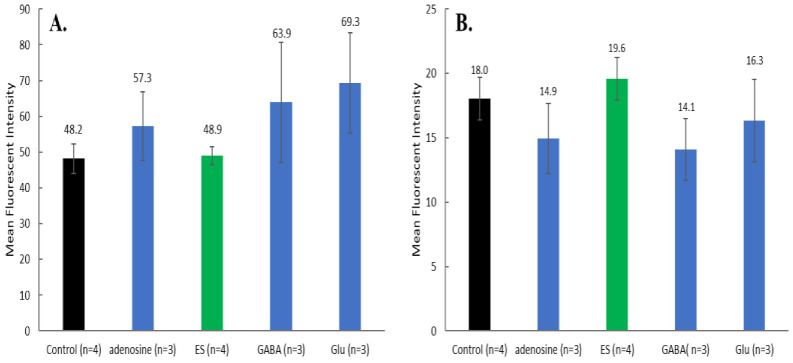
(**A**) Mean fluorescent intensity ± SD of Fluo-4 AM (Ca^2+^ probe) averaged by condition group. All differences in mean intensities are significant except ES vs. Control by ANOVA with Tukey post hoc test (*p* < 0.05). (**B**) Mean fluorescent intensity ± SD of MQAE (Cl^−^) averaged by condition group. No statistical difference was found between any of the groups.

**Table 1 brainsci-12-01504-t001:** Stock and final concentrations of neurotransmitters used in experimentation.

Neurotransmitter	Stock Solution	Final Well Concentration
Glutamate	100 mM	10 mM
Adenosine	2 μM	200 nM
D-serine	2 mM	200 μM
GABA	1 mM	100 μM

## Data Availability

Data are contained within this article.

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
