# Peer review of "Development of a C6 Glioma Cell Model System to Assess Effects of Cathodic Passively Balanced Electrical Stimulation on Responses to Neurotransmitters: Implications for Modulation of Intracellular Nitric Oxide, Chloride, and Calcium Ions"

_brainsci, 2022, doi:10.3390/brainsci12111504_

Round 1

Reviewer 1 Report

Nel paper Development of a C6 Glioma Cell Model System to Assess Effects of Cathodic Passively Balanced Electrical Stimulation on Responses to Neurotransmitters: Implications for Modulation of Intracellular Nitric Oxide, Chloride, and Calcium Ions the authors focused on the development of an astrocyte cell model system (C6 glioma) for the assessment of molecular changes in response to cathodic passively balanced pulsed electrical stimulation at a rate of 50 Hz.  Using specific fluorescent probes and stimulating with specific selected neurotransmitters (glutamate, adenosine, D-serine, and γ-aminobutyric acid) after electrical stimulation of C6 cell  they evaluate the calcium ion /chloride changing.

The paper is well described but I have many doubts:

1)The authors use a tumor glial cell model that may have a different response to electrical stimulation. Do the authors have evidence of stimulation on normal glia cells?

2) Pain can also be potentiated by growth factors such as BDNF and bFGF that are produced by glia to protect neurons. Do the authors have evidence of  changing levels of BDNF and bFGF after C6 stimulation?

The study is interesting but should be reviewed as a description of the variations induced by electrical stimulation of 50HZ of C6 glioblastoma cells. I don't think it can represent a real model of the intracellular dynamics of pain

Author Response

please find it in the attachment

Reviewer 2 Report

In the present study, Platt et al. used passively balanced cathodic pulses (50 Hz) and monitored nitric oxide (NO), chloride (Cl-), and calcium ions (Ca2+) in C6 glioma cells under the presence of different neurotransmitters compared to the cathodic stimulation alone. The study has solid methodological potential for evaluating cell activity in vitro; however, the work needs significant improvement. 

There are several major concerns:

1.         Glutamate (Glu, 10 mM) concentration utilized in this study is significantly higher than the physiological Glu concentration. Glu (from 5 to 100 uM) is usually used to activate the ionotropic Glu- activated ion channels. The prolonged application of Glu at a high concentration induces desensitization of the ionotropic AMPA and NMDA receptors; therefore, the data interpretation or the experiments need to be redesigned. Also, 10 mM of Glu could be toxic for cells; the cell-viability data will be helpful for data interpretation. 

2.         It will be nice to provide transcriptomic data confirming the expression of the Glu- activated ionotropic receptors in C6 cells. The observed effects could be related to metabotropic glutamate receptors. 

3. Receptor-specific antagonists could be utilized to improve data interpretation. 

4.         The calibration data should be provided in the experiments with calcium-sensitive dye (Fluo-4 AM). Calcium ionophore ionomycin could be utilized as a control in some experiments with Fluo-4 AM. Please include this information in the method section.  

5.         The transcriptomic or proteomic data will be helpful for confirming the expression of GABAa and other receptors in C6 cells (or references to the existing data).

6.         Calcium oscillations and calcium transient signals should be observed in a time-dependent manner and could be included in the data interpretation (or excluded from data analysis). Please provide information related to calcium oscillation and calcium transients.  

Minor:

1.         The Y axis is missing in the graphs presented in Fig. 3, Fig. 4, Fig. 5, Fig. 7, Fig. 8, Fig. 10, and Fig. 11.

Author Response

please find it in the attachment

Reviewer 3 Report

In their manuscript entitled “Development of a C6 Glioma Cell Model System to Assess Effects of Cathodic Passively Balanced Electrical Stimulation on Responses to Neurotransmitters: Implications for Modulation of Intracellular Nitric Oxide, Chloride, and Calcium Ions”, Platt et al. sought to investigate the potential effects of cathodic passively balanced pulsed electrical stimulation (at 50 Hz) on the production and maintenance of specific chemical species in an astrocyte cell model system (C6 glioma). This in vitro model system may be used to study cellular mechanisms involved in the development and maintenance of chronic pain.

Overall, this article describes an interesting scientific idea. Elucidating mechanisms of novel treatment strategies in the management of chronic pain constitutes a clear medical need in neurology. However, there are several concerns/issues in the manuscript the authors must address before the article may be considered for publication:

Major concerns:

1.     A rationale as to why the selected cell line was chosen has to be provided. Why was a glioma cell line used as the single cell model for this study? Can the authors proof that this tumor cell line does not differ from non-cancerous glial cells, which are presumably the cells of interest for this research, regarding the herein investigated parameters?

2.     Why did the authors decide to use an epifluorescent microscope instead of using confocal microscopy, which, generally speaking, offers several distinct advantages over traditional widefield fluorescence microscopy.

3.     Statistics for figure 1-5 are missing and should be included.

4.     The high number of individual figures is confusing. I strongly recommend summarizing figures of related experiments and making sub-figures.

5.     The discussion in its current form has substantial shortcomings and needs to be re-written from the ground up. I kindly refer to common resources in print and on the internet on how to structure the discussion section of a manuscript.

Instead of regurgitating the results, the authors should 1) briefly summarize their main findings, 2) discuss their findings in the context of the published literature, 3) highlight limitations and future directions of their study.

These and other questions should be discussed:

·      Specifically, with respect to their findings, the authors say that "Spectroscopy measurements, however, may be a preferred method of monitoring changes." - Why? Important point of discussion. Elaborate on this statement.

·      How exactly is the described methodology helpful for in vivo studies in animals and humans (except for the fact that it is quicker and less expensive)?

·      What are the limitations of the study?

·      What are the future directions of this research?

6.     Similar to the discussion, the "conclusion" needs to be rewritten from scratch in order to actually be considered as a conclusion section.

Minor concerns:

1.     The abstract needs to include a description of results. The conclusion portion should be shortened.

2.     All figures need explanatory legends to enable them to stand independent of the main text.

3.     Figure legends need to be formatted as such.

4.     The description of why certain neurotransmitters were used in this study (e.g., lines 63-64, 87, etc.) should be moved to the methods section 2.2., for example after line 137.

5.     Lines 156-157: This sentence needs to be referenced.

6.     Line 198: The authors write “with exposure values set to the control well level”. What does this mean? Clarify, so experiment could be repeated by independent researcher.

7.     Line 258: “Figure 1)” should read Figure 3) here.

8.     Line 263ff: What about the percent of cells fluorescent with this probe? Like figure 3...

9.     P values should be included in the main text. In the methods section, it should be described what Asterisks and daggers indicate.

10.  Figures 6 and 9: Make scale bar bigger, label A-D.

11.  Lines 307-310: This sentences, and similar passages throughout the results section, should be moved to the discussion (limitations of the study).

Author Response

please find it in the attachment

Round 2

Reviewer 2 Report

The revised manuscript provides a significant improvement in data interpretation and presentation. It exhibits methodological potential for evaluating cell activity in future experiments in vitro; therefore, the manuscript is of great interest to readers. I recommend manuscript for publication in Brain Sciences.

Reviewer 3 Report

I appreciate the authors' revision and recommend publication of the manuscript in its current form in Brain Sciences.